# Unsupervised Learning of Video Representations via Dense Trajectory Clustering

Pavel Tokmakov[1], Martial Hebert[1], and Cordelia Schmid[2]

[1] Carnegie Mellon University
[2] Inria

**Abstract.** This paper addresses the task of unsupervised learning of representations for action recognition in videos. Previous works proposed to utilize future prediction, or other domain-specific objectives to train a network, but achieved only limited success. In contrast, in the relevant field of image representation learning, simpler, discrimination-based methods have recently bridged the gap to fully-supervised performance. We first propose to adapt two top performing objectives in this class - instance recognition and local aggregation, to the video domain. In particular, the latter approach iterates between clustering the videos in the feature space of a network and updating it to respect the cluster with a non-parametric classification loss. We observe promising performance, but qualitative analysis shows that the learned representations fail to capture motion patterns, grouping the videos based on appearance. To mitigate this issue, we turn to the heuristic-based IDT descriptors, that were manually designed to encode motion patterns in videos. We form the clusters in the IDT space, using these descriptors as a an unsupervised prior in the iterative local aggregation algorithm. Our experiments demonstrates that this approach outperform prior work on UCF101 and HMDB51 action recognition benchmarks. We also qualitatively analyze the learned representations and show that they successfully capture video dynamics.

**Keywords:** unsupervised representation learning, action recognition

## 1   Introduction

The research on self-supervised learning of image representation has recently experienced a major breakthrough. Early approaches carefully designed objective functions to capture properties that the authors believed would result in learning rich representations [6,29,11,48]. For instance, Doersch et al. [6] proposed to predict relative positions of two patches in an image, and Zhang et al. [48] trained a network to colorize images. However, they have achieved only limited success. The methods that have brought the performance of self-supervised image representations close to those learned in a fully-supervised way, rely on a different principle instead. They use the standard cross-entropy loss and either treat each image as an individual class [8,46,30], or switch between clustering images in the feature space of the network, and updating the model to classify them into

clusters [1,49]. The resulting representations effectively capture discriminative image cues without having to manually separate images into categories.

Self-supervised feature learning for videos has so far mostly relied on manually designed objective functions. While some works adopted their objectives directly from the image-based methods, such as predicting video rotation [19], or relative position of space-time patches [20], others utilize video-specific cues, such as predicting feature representations of video patches in future frames [13]. Very recently, Sun et al. [35], have proposed a variant of the instance classification objective for videos.

In this work we first investigating whether the recent, classification-based objectives proposed for image representation learning can be applied to videos. We introduce a video variant of the non-parametric Instance Recognition approach of Wu et al., [46] (Video IR). It simply treats each video as its own class and trains a 3D ConvNet [36,14] to discriminate between the videos. We observe that this naive approach is already competitive with prior work in the video domain.

To further improve the results, we capitalize on the observation of Zhuang et al. [49] that embedding semantically similar instances close to each other in feature space is equally important to being able to discriminate between any two of them. We adapt their Local Aggregation approach to videos (Video LA). As shown in the top part of Figure 1, this method first encodes a video using a 3D ConvNet, and the resulting embeddings are clustered with K-means. A non-parametric clustering loss proposed in [49] is then used to update the network and the algorithm is iterated in an Expectation-Maximization framework. This approach results in an improvement over Video IR, but the gap between the two objectives remains smaller than in the image domain.

We identify the reasons behind this phenomenon, by examining the video clusters discovered by the algorithm. Our analysis shows that they mainly capture appearance cues, such as scene category, and tend to ignore the temporal information, which is crucial for the downstream task of action recognition. For instance, as shown in the top right corner of Figure 1, videos with similar background, but different activities are embedded closer than examples of the same action. This is not surprising, since appearance cues are both dominant in the data itself, and are better reflected in the 3D ConvNet architecture.

To mitigate this issue, we turn to the heuristic-based video representations of the past. Improved Dense Trajectories (IDT) [42] were the state-of-the-art approach for action recognition in the pre-deep learning era, and remained competitive on some datasets until very recently. The idea behind IDT is to manually encode the cues in videos that help to discriminate between human actions. To this end, individual pixels are first tracked with optical flow, and heuristics-based descriptors [4,5,41] are aggregated along the trajectories to encode both appearance and motion cues.

In this work, we propose to transfer the notion of similarity between videos encoded in IDTs to 3D ConvNets via non-parametric clustering. To this end, we first compute IDT descriptors for a collection of unlabeled videos. We then cluster these videos in the resulting features space and use the non-parametric

classification objective of [49] to train a 3D ConvNet to respect the discovered clusters (bottom part of Figure 1). The network is first trained until convergence using the fixed IDT clusters, and then finetuned in the joint IDT and 3D ConvNet space with the iterative Video LA approach. The resulting representation outperforms the baselines described above by a significant margin. We also qualitatively analyze the clusters and find that they effectively capture motion information.

Following prior work [13,19,35], we use the large-scale Kinetics [2] dataset for self-supervised pretraining, ignoring the labels. The learned representations are evaluated by finetuning on UCF101 [33] and HMDB51 [23] action recognition benchmarks. To gain a better insight into the properties of the representations, we additionally provide an in-depth qualitative and quantitative analysis of the proposed approach.

## 2   Related work

In this section, we first briefly review previous work on image-based unsupervised representation learning. We then discuss various approaches to video modeling, and conclude by presenting relevant video representation learning methods.

**Image representation** learning from unlabeled data is a well explored topic. Due to space limitations, we will only review the most relevant approaches here. The earliest methods were built around auto-encoder architectures: one network is trained to compress an image into a vector in such a way, that another network is able to reconstruct the original image from the encoding [18,24,21,7,12]. In practice, however, the success of generative methods in discriminative representation learning has been limited.

Until very recently, manually designing self-supervised objectives has been the the dominant paradigm. For example, Doersch et al. [6] and Noroozi and Favaro [29] predict relative positions of patches in an image, Zhang et al. [48] learn to colorize images, and Gidaris et al. [11] learn to recognize image rotations. While these methods have shown some performance improvements compared to random network initialization, they remain significantly below a fully-supervised baseline. The most recent methods, instead of designing specialized objective functions, propose to use the standard cross-entropy loss and either treat every image as its own class [8,30,46], or switch between clustering the examples in the feature space of the network and updating the network with a classification loss to respect the clusters [1,49]. These methods exploit the structural similarity between semantically similar images, to automatically learn a semantic image embedding. In this paper we adapt the methods of Wu et al. [46] and Zhuang et al. [49] to the video domain, but demonstrate that they do not perform as well due to the structural priors being less strong in videos. We then introduce explicit prior in the form of IDT descriptors and show this indeed improves performance.

**Video modeling** has traditionally been approached with heuristics-based methods. Most notably, Dense Trajectories (DT) [41] sample points in frames and track them with optical flow. Then appearance and motion descriptors are

extracted along each track and encoded into a single vector. The discriminative ability of DT descriptors was later improved in [42] by suppressing camera motion with the help of a human detector, and removing trajectories that fall into background regions. The resulting representation focuses on relevant regions in videos (humans and objects in motion) and encodes both their appearance and motion patterns.

More recently, the success of end-to-end trainable CNN representation has been extended to the video domain. Simonyan et al. [32] proposed to directly train 2D CNNs for action recognition, fusing several frames at the first layer of the network. Their approach, however, had a very limited capacity for modeling temporal information. This issue was later addressed in [36] by extending the 2D convolution operation in time. Introduction of the large scale Kinetcis dataset for action recognition [2] was a major step forward for 3D CNNs. Pretrained on this dataset, they were finally able to outperform the traditional, heuristic-based representations. Several variants of 3D ConvNet architectures have been proposed since, to improve performance and efficiency [2,14,47]. In this work, we demonstrate how the IDT descriptors can be used to improve unsupervised learning of 3D ConvNet representations.

**Video representation** learning from unlabeled data is a less explored topic. This is largely because the community has only recently converged upon the 3D ConvNets as thr standard architecture. Early methods used recurrent networks, or 2D CNNs, and relied on future-prediction [34], as well as various manually designed objectives [28,27,25,10,9]. In particular, several works utilized temporal consistency between consecutive frames as a learning signal [27,25,28], whereas Gan et al. [10] used geometric cues, and Fernando et al. [9] proposed the odd-one-out objective function.

With 3D ConvNets, generative architectures [20,38], as well as some self-supervised objectives have been explored [19,20,43]. For example, Jing et al. [19] train a model to predict video rotation, Kim et al. [20] use relative spatio-temporal patch location prediction as an objective, and Wang et al. [43] regress motion and appearance statistics. In another line of work, future frame colorization was explored as a self-supervision signal [39]. Recently, Han et al. [13] proposed to predict feature representations of video patches in future frames. Most similarly, Sun et al. [35] use a variant of the instance discrimination loss. In this work, we demonstrate that simply adapting instance discrimination [46] and local aggregation [49] objectives from the image to the video domain already achieves competitive results, and augmenting local aggregation with IDT priors further improves the results, outperforming the state-of-the-art.

## 3   Method

Our goal is to learn an embedding function $f_{\boldsymbol{\theta}}$ that maps videos $V = \{v_1, v_2, ..., v_N\}$ into compact descriptors $f_{\boldsymbol{\theta}}(v_i) = \boldsymbol{d}_i$ in such a way, that they can be discriminated based on human actions, using unlabeled videos. For instance, as shown in Figure 1, we want the two videos of people to doing handstands to be close to

each other in the embedding space, and well separated from the video of a person training a dog. Below, we first introduce the two objective functions used in our work - instance recognition [46] and local aggregation [49], and then describe our approach of using IDT [42] descriptors as unsupervised priors in non-parametric clustering.

### 3.1   Video instance recognition

This objective is based on the intuition that the best way to learn a discriminative representation is to use a discriminative loss. And, in the absence of supervised class labels, treating each instance as a distinct class of its own is a natural surrogate.

Using the standard softmax classification criterion, the probability of every video $v$ with the feature $\boldsymbol{d}$ belonging to its own class $i$ is expressed as:

$$P(i|\boldsymbol{d}) = \frac{\exp(\boldsymbol{w}_i^T \boldsymbol{d})}{\sum_{j=1}^{N} \exp(\boldsymbol{w}_j^T \boldsymbol{d})}, \tag{1}$$

where $\boldsymbol{w}_j$ is the weight vector of the $j$'th classifier. In this case, however, every class contains only a single example, thus $\boldsymbol{w}_j$ can be directly replaced with $\boldsymbol{d}_j$. The authors of [46] then propose the following formulation of the class probability:

$$P(i|\boldsymbol{d}) = \frac{\exp(\boldsymbol{d}_i^T \boldsymbol{d}/\tau)}{\sum_{j=1}^{N} \exp(\boldsymbol{d}_j^T \boldsymbol{d}/\tau)}, \tag{2}$$

where $\tau$ is a temperature parameter that controls the concentration level of the distribution, and helps convergence [40,17]. The final learning objective is the standard negative log likelihood over the training set. Recall that training is done in batches, thus a memory bank of encodings $D = \{\boldsymbol{d}_1, \boldsymbol{d}_2, ..., \boldsymbol{d}_N\}$ has to be maintained to compute Equation 2.

### 3.2   Video local aggregation

While being able to separate any two instances is a key property for an image or video embedding space, another, complementary and equally desirable property is minimizing the distance between semantically similar instances. To this end, Zhuang et al. [49] proposed to use clusters of instances instead of individual examples as class surrogates. We adapt their approach to the video domain, and briefly describe it below.

Firstly, the video embedding vectors $\boldsymbol{d}_1, \boldsymbol{d}_2, ..., \boldsymbol{d}_N$ are grouped into $K$ clusters $G = \{G_1, G_2, .., G_K\}$ using K-means. The embedding function $f_{\boldsymbol{\theta}}$ is then updated to respect the cluster, using the non-parametric clustering objective proposed in [49], and the two steps are iterated in an EM-framework. In particular, for every instance $v_i$ together with its embedding $\boldsymbol{d}_i$, two sets of neighbours are identified: close neighbours $\boldsymbol{C}_i$ (shown with a dashed circle in Figure 1) and background neighbours $\boldsymbol{B}_i$. Intuitively, close neighbours are those examples that

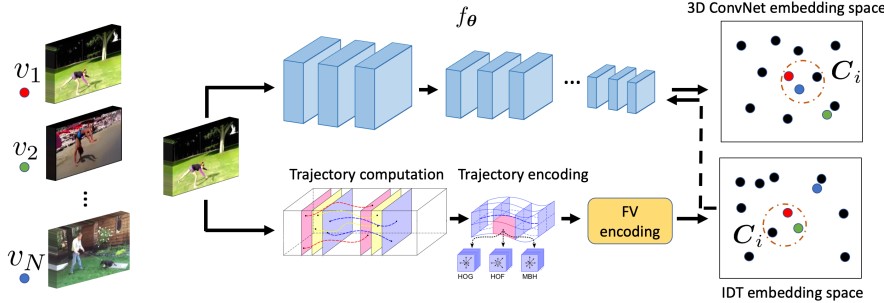

**Fig. 1.** Our approach for unsupervised representation learning from videos. Directly applying a non-parametric clustering objective results in a representation that groups videos based on appearance (top right corner). To mitigate this issue, we propose to first cluster videos in the space of IDT descriptors (bottom right corner), which results in a grouping that better reflects video dynamics. We then apply the non-parametric clustering loss to transfer the properties of this embedding to a 3D ConvNet.

fall into the same cluster as $v_i$ and background neighbors are simply those that have a small distance to $d_i$ in the feature space (they include both close neighbors and hard negative examples). Please see [49] for more details on how $C_i$ and $B_i$ are constructed.

The objective is then to minimize the distance between $d_i$ and its close neighbours (instances in the same cluster), while maximizing the distance to those background neighbors that are not in $C_i$ (hard negatives). The authors formulate this objective in a probabilistic way as minimizing the negative log likelihood of $d_i$ being recognized as a close neighbor, given that it is recognized as a background neighbor:

$$L(C_i, B_i | d_i, \theta) = - \log \frac{P(C_i \cap B_i | d_i)}{P(B_i | d_i)}, \tag{3}$$

where the probability of $d$ being a member of a set $A$ is defined as:

$$P(A | d) = \sum_{i \in A} P(i | d), \tag{4}$$

and the definition of $P(i|d)$ is adapted from Equation 2. Despite the involved formulation, one can see that this objective does exactly what it is intended to do - minimizes the distance between examples inside a cluster and maximize it between those belonging to different clusters in a non-parametric way.

Intuitively, the Local Aggregation objective relies on the structural similarity between semantically similar images, together with deep image prior in CNN architectures [37], to form meaningful clusters in the embedding space. In videos, however, both structural and architectural priors are less strong. Indeed, pixels that are close to each other in the spatio-temporal volume of a video are not always strongly correlated due to the presence of object and camera and motion. On the architecture side, 3D ConvNets are also worse at capturing spatio-temoral

patterns, compared to CNNs at capturing spatial patterns. To mitigate this lack of implicit priors, we propose to introduce an explicit one in the form of IDT descriptors.

### 3.3   IDT descriptors as priors for video representation learning

While state-of-the-art architectures for action recognition [36,2,14] simply extend 2D CNN filters into the temporal dimension, treating videos as spatio-temporal cuboids of pixels, classical approaches [41,42] explicitly identified and encoded spatio-temporal interest points that are rich in motion patterns relevant to action classification. In our experiments, we use the original implementation of IDT [42] to compute video descriptors for unlabeled videos (shown in the lower part of Figure 1). We supply the IDT extractor with human detection form the state-of-the-art Mask-RCNN [16] model trained on MS COCO [26] for improved camera stabilization (see [42] for details).

   This method, however, produces thousands of descriptors $x \in \mathcal{X}$ per video. To encode them into a compact vector we follow prior work [42,44] and first apply PCA to reduce the dimensionality of each individual trajectory descriptor $x_i$. We then utilize Fisher vector coding [31], which is based on a Gaussian Mixture Model (GMM) with K components $G(w_k, \boldsymbol{\mu}_k, \boldsymbol{\sigma}_k)$, parameterized by mixing probability, mean, and diagonal standard deviation. The encoding for a trajectory descriptor $x$ is then computed by stacking the derivatives of each components of the GMM with respect to mean and variance:

$$\phi_k^*(\boldsymbol{x}) = \frac{p(\boldsymbol{\mu}_k|\boldsymbol{x})}{\sqrt{w_k}}[\phi_k(\boldsymbol{x}), \frac{\phi_k^{'}(\boldsymbol{x})}{\sqrt{2}}], \tag{5}$$

where the first- and second-order features $\phi_k, \phi_k^{'} \in R^D$ are defined as:

$$\phi_k(\boldsymbol{x}) = \frac{(\boldsymbol{x} - \boldsymbol{\mu_k})}{\boldsymbol{\sigma}_k}, \phi_k^{'}(\boldsymbol{x}) = \phi_k(\boldsymbol{x})^2 - 1, \tag{6}$$

thus, the resulting Fisher vector encoding $\phi(\boldsymbol{x}) = [\phi_1^*(\boldsymbol{x}), \phi_2^*(\boldsymbol{x}), ..., \phi_k^*(\boldsymbol{x})]$ is of dimensionality $2KD$. To obtain the video-level descriptor $\boldsymbol{\psi}$, individual trajectory encodings are averaged $\boldsymbol{\psi} = avg_{\boldsymbol{x} \in \mathcal{X}}\phi(\boldsymbol{x})$, and power- [22] and l2-normalization are applied. Finally, to further reduce dimensionality, count sketching [45] is used: $p(\boldsymbol{\psi}) = \boldsymbol{P}\boldsymbol{\psi}$, where $\boldsymbol{P}$ is the sketch projection matrix (see [45] for details).

   The resulting encoding $p(\boldsymbol{\psi})$ is a 2000-dimensional vector, providing a compact representation of a video, which captures discriminative motion and appearance information. Importantly, it is completely unsupervised. Both the PCA projection and the parameters of the Gaussian mixture model are estimated using a random sample of trajectory encodings, and matrix $\mathbf{P}$ is selected at random as well.

   To transfer the cues encoded in IDTs descriptors to a 3D ConvNet, we first cluster the videos in the $p(\boldsymbol{\psi})$ space with K-means, to obtain the clusters $G$. We then use $G$ to compute the sets of neighborhoods $(\boldsymbol{C}_i, \boldsymbol{B}_i)$ for each video $v_i$ in an unlabeled collection (shown in the bottom right corner on Figure 1), and

apply the objective in Equation 3 to train the network. This forces the learned representation to capture the motion patterns that dominate the IDT space (note that IDTs encode appearance cues as well in the form of HOG descriptors).

Finally, we construct a joint space of IDT and 3D ConvNet representations by concatenating the vectors $d$ and $p(\psi)$ for each video. We further finetune the network in this joint space for a few epochs by concatenating the respective feature representations before clustering. This step allows the model to capitalize on appearance cues encoded by the the expressive 3D ConvNet architecture. We analyze the resulting model quantitatively and qualitatively, and find that it both outperforms the state-of-the-art, and is better at capturing motion information.

## 4    Experiments

### 4.1    Datasets and evaluation

We use the Kinetics [2] dataset for unsupervised representation learning and evaluate the learned models on UCF101 [33] and HMDB51 [23] in a fully-supervised regime. Below, we describe each dataset in more detail.

**Kinetics** is a large-scale, action classification dataset collected by querying videos on YouTube. We use the training set of Kinetics-400, which contains 235 000 videos, for most of the experiments in the paper, but additionally report results using fewer as well as more videos in Section 4.6. Note that we do not use any annotations provided in Kinetics.

**UCF101** is a classic dataset for human action recognition, which consists of 13,320 videos, covering 101 action classes. It is much smaller than Kinetics, and 3D ConvNets fail to outperform heuristic-based methods on it without fully-supervised pretraining on larger datasets. Following prior work [19,13], we use UCF101 to evaluate the quality of representations learned on Kinetics in an unsupervised way via transfer learning. In addition to using the full training set of UCF101, we report few-shot learning results to gain more insight into the learned representations in the supplementary material. We use the first split of the dataset for ablation analysis, and report results averaged over all splits when comparing to prior work.

**HMDB51** is another benchmark for action recognition, which consists of 6,770 videos, collected from movies, and split into 51 categories. Due to the small size of the training set, it, poses an even larger challenge for learning-based methods. As with UCF101, we report ablation results on the first split, and use the results averaged over all splits for comparison to prior work.

Following standard protocol, we report classification accuracy as the main evaluation criteria on UCF101 and HMDB51. However, this makes direct comparison between different approaches difficult, due to the differences in network architectures. Thus, whenever possible, we additionally report the fraction of the fully-supervised performance for the same architecture.

## 4.2   Implementation details

**Self-supervised objectives** We study three self-supervised objective functions: Video Instance Recognition (Video IR), Video Local Aggregation (Video LA) and Video Local Aggregation with IDT prior. For Video IR we follow the setting of [46] and set $\tau$ in Equation 2 to 0.07. We use 4096 negative samples for approximating the denominator of Equation 2.

In addition to the parameters described above, Local Aggregation requires choosing the number of clusters $K$, as well as the number of runs of K-means that are combined for robustness. The authors of [49] do not provide clear guidelines on selecting these hyperparameters, so we choose to take the values used in their ImageNet experiments and decrease them proportionally to the size of Kinetics. As a result, we set $K$ to 6000 and the number of clusterings to 3. We validate the importance of this choice ,and provide the implementation details for the IDT computation in the supplementary material.

**Network architecture and optimization** Following most of the prior work, we use a 3D ResNet18 architecture [14] in all the experiments, but also report results with deeper variants in the supplementary material. The embedding dimension for self-supervised objectives is set to 128, as in [49]. We use SGD with momentum to train the networks, and apply multi-scale, random spatio-temporal cropping for data augmentation, with exactly the same setting as in [14]. We also perform the standard mean subtraction. All the models are trained on 16 frames clips of spatial resolution of $112 \times 112$, unless stated otherwise.

During self-supervised learning we follow the setting of [49] and set the learning rate to 0.03, and momentum to 0.9, with batch size of 256. All the models are trained for 200 epoch, and the learning rate is dropped by a factor 0.1 at epochs 160 and 190. As in [49], we initialize the LA models with 40 epoch of IR pretraining.

## 4.3   Analysis of self-supervised objectives

We begin by comparing different variants of self-supervised objectives described in Section 3. They are used to learn a representation on Kinetics-400 in a self-supervised way, and the resulting models are transferred to UCF101 and HMDB51. We additionally evaluate two baselines - Supervised, which is pretrained on Kinetics using ground-truth labels, and Scratch, which is initialized with random weights. The results are reported in Table 1.

Firstly, we observe that supervised pretraining is indeed crucial for achieving top performance on both datasets, with the variant trained from scratch reaching only 50.2% and 30.3% of the accuracy of the fully supervised model on UCF101 and HMDB51 respectively. The gap is especially large on HMDB51, due to the small size of the dataset. Using the video variant of the Instance Recognition objective (Video IR in the table), however, results in a 27.6% accuracy improvement on UCF101 and 22.8% HMDB51, reaching 82.9% and 70.7% of the supervised accuracy respectively. Notice that this simple method already outperforms some of the approaches proposed in prior works [19,13,20].

**Table 1.** Comparison between variants of unsupervised learning objective using classification accuracy and fraction of fully supervised performance on the first split of UCF101 and HMDB51. All models use a 3D ResNet18 backbone, and take 16 frames with resolution of $112 \times 112$ as input. Video LA with IDT prior consistently outperforms other objectives, with improvements on HMDB51 being especially significant.

| Method | UCF101 | | HMDB51 | |
|---|---|---|---|---|
| | Accuracy | % sup. | Accuracy | % sup. |
| Scratch [14] | 42.4 | 50.2 | 17.1 | 30.3 |
| Video IR | 70.0 | 82.9 | 39.9 | 70.7 |
| Video LA | 71.4 | 84.6 | 41.7 | 73.9 |
| Video LA + IDT prior | **72.8** | **86.3** | **44.0** | **78.0** |
| Supervised [14] | 84.4 | 100 | 56.4 | 100 |

Next, we can see that the Local Aggregation objective (Video LA in the table) further improves the results, reaching 84.6% and 73.9% of the fully-supervised performance on UCF101 and HMDB51 respectively. This shows that despite the higher-dimensionality of the video data, this method is still able to discover meaningful clusters in an unsupervised way. However, the gap to the IR objective is smaller than in the image domain [49].

Finally, our full method, which uses IDT descriptors as an unsupervised prior when clustering the videos (Video LA + IDT prior in the table), is indeed able to further boost the performance, reaching 86.3% and 78.0% of fully supervised performance on the two datasets. The improvement over Video LA is especially significant on HMDB51. We explain this by the fact that categories in UCF101 are largely explainable by appearance, thus the benefits of better modeling the temporal information are limited on this dataset. In contrast, on HMDB51 capturing scene dynamics is crucial for accurate classification.

### 4.4   Qualitative analysis of the representations

To gain further insight into the effect of our IDT prior on representation learning, we now visualize some of the clusters discovered by the vanilla LA, and the variant with the prior in Figures 2 and 3 respectively. Firstly, we observe that, in the absence of external constraints LA defaults to using appearance, and primarily scene information to cluster the videos. For instance, the first cluster (top left corner) corresponds to swimming pools, the one on the top right seems to focus on grass, and the two clusters in the bottom row capture vehicles and backyards, irrespective of the actual scene dynamics. This is not surprising, since appearance cues are both more dominant in the data itself, and are better reflected by the 3D ConvNet architecture.

In contrast, the model learned with IDT prior is better at capturing motion cues. For example, the cluster in the top left corner of Figure 3 is characterized by forward-backward hand motion, such as observed during cleaning or barbecuing. The cluster in the top-right captures humans spinning or rotating. The bottom left cluster mostly contains videos with very fast actor motion, and the one in the bottom right closely corresponds to the action 'riding'.

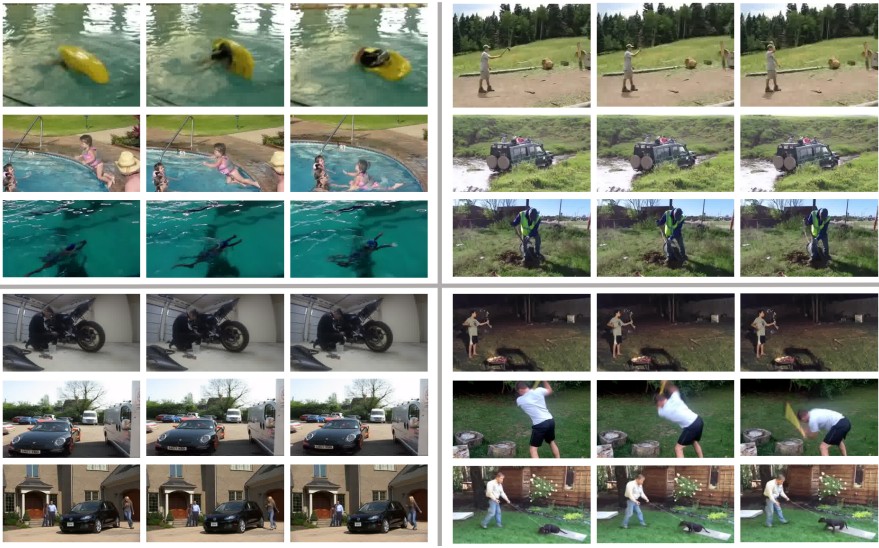

**Fig. 2.** Visualization of the clusters discovered by the Video LA objective without IDT prior. This variant groups videos in the space of a 3D ConvNet. As a results, the clusters are primarily defined by the appearance, grouping swimming pools, grass fields, vehicles, and backyards. The activity happening in the videos does not seem to play a significant role.

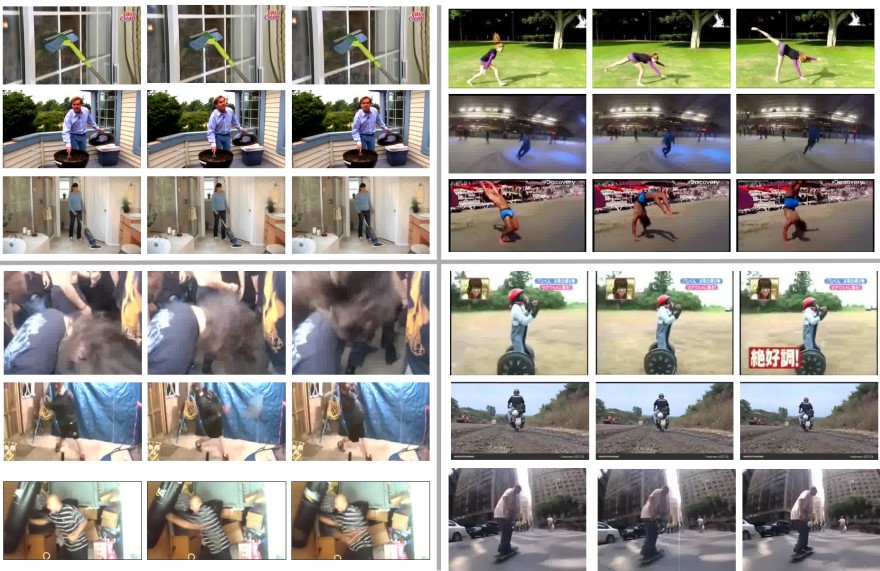

**Fig. 3.** Visualization of the clusters discovered by variant of Video LA objective that uses IDT prior. In contrast to the examples above, the videos are mainly grouped by motion properties, such as forward-backward hand motion, person rotation, fast person motion, and 'riding' action.

**Table 2.** Evaluation of the effect of clip length on various objectives on the first split of UCF101 and HMDB51 using classification accuracy. All models use a 3D ResNet18 backbone, and take frames with resolution of $112 \times 112$ as input. Both self-supervised and fully-supervised variants benefit from longer sequences, but the model trained from scratch is not able to capitalize on more information.

| Method | UCF101 | | | HMDB51 | | |
|---|---|---|---|---|---|---|
| | 16-fr | 32-fr | 64-fr | 16-fr | 32-fr | 64-fr |
| Scratch | 42.4 | 44.9 | 45.3 | 17.1 | 18.0 | 17.4 |
| Video LA | 71.4 | 75.0 | 79.4 | 41.7 | 43.1 | 48.9 |
| Video LA + IDT prior | **72.8** | **76.3** | **81.5** | **44.0** | **44.7** | **49.6** |
| Supervised | 84.4 | 87.0 | 91.2 | 56.4 | 63.1 | 67.5 |

Importantly, neither set of clusters is perfectly aligned with the definition of actions in popular computer vision dataset. For instance, despite having a clear motion-based interpretation, the top left cluster in Figure 3 combines Kinetics categories 'cleaning window', 'cleaning floor', and 'barbecuing'. Indeed, the actions vocabulary used in the literature is defined by a complex combination of actor's motion and scene appearance, making automatic discovery of well-aligned clusters challenging, and partially explaining the remaining gap between clustering-based methods and fully-supervised pretraining.

### 4.5   Learning long-term temporal dependencies

Next, we experiment with applying our Video LA objective with IDT prior over longer clips. Recall that this approach attempts to capture the notion of similarity between the videos encoded in the IDT descriptors that are computed over the whole video. The model reported so far, however, only takes 16-frame clips as input, which makes the objective highly ambiguous. In Table 2 we evaluate networks trained using 32- and 64-frame long clips instead, reporting results on UCF101 and HMDB51.

We observe that, as expected, performance of our approach ('Video LA + IDT' in the table) increases with more temporal information, but the improvement is non-linear, and our model is indeed able to better capture long-term motion cues when trained using longer clips. Similar improvements are observed for the plain Video LA objective, but our approach still shows top performance. Supervised model is also able to capitalize on longer videos, but on UCF101 the improvements are lower than seen by our approach (6.8% for the supervised model, compared to 8.7% for ours). Interestingly, the model trained from scratch does not benefit from longer videos as much as self-supervised or supervised variants. In particular, on HMDB51 its performance improves by about 1-2% with 32 frames, but actually decreases with 64. We attribute this to the fact that using longer clips lowers the diversity of the training set. These results further demonstrate the importance of model pretraining for video understanding.

### 4.6    Effect of the number of videos

So far, we have reported all the results using 235 000 videos from Kinetics-400 [2]. We now train the model with our final objective (Video LA with IDT prior) using a varying number of videos to study the effect of the dataset size on the quality of representations. In particular, we subsample the training set to 185 000 and 135 000 examples at random to see whether smaller datasets can be used for representation learning. We also add the videos from Kinetics-600 to see if our method scales to larger video collections. We use the 3D ResNet18 architecture with 16-frames long clips and input resolution of $112 \times 112$ in all experiments, and report results on the first split of UCF101 and HMDB51 in Figure 4.

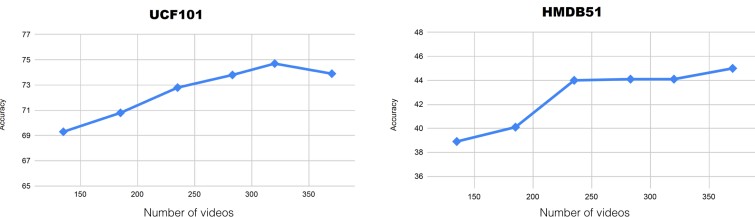

**Fig. 4.** Varying the number of Kinetics videos when training a 3D ConvNet with the 'Video LA with IDT prior' objective. Using more data for unsupervised pretraining results in better representations, as evident form transfer learning results on the first split of UCF101 and HMDB51 (reported using classification accuracy).

Firstly, we observe that useful representations can be learned with as few 135 000 videos. However, using more data results in improved performance on both datasets. On UCF101 the improvements are mostly linear, but accuracy drops somewhat for the largest training set (370 000 videos). We attribute this to the randomness in training and hypothesize that further improvements can be achieved with more data. On HMDB51 accuracy seems to plateau after 235 000 videos, but improves with 370 000. We will use the model trained on the largest available dataset for comparison to the state-of-the-art in the next section.

### 4.7    Comparison to the state-of-the-art

Finally, we compare our approach (Video LA with IDT prior) to the state-of-the-art in Table 3. To fairly compare results achieved by methods with different network architectures, we use the fraction of fully supervised performance as an additional metric, whenever this information is available. To make the table size manageable, we only report approaches that use 3D ConvNets pretrained on Kinetics. These, however, cover all the top performing methods in the literature.

Firstly, we observe that our principled approach is indeed a lot more effective that manually designed objectives used in PMAS [43], or 3D-Puzzle [19], confirming the effectiveness of clustering-based training. The improvements are especially large on HMDB, which is, as we have shown previously, can be attributed to the

**Table 3.** Comparison to the state-of-the-art using accuracy and fraction of the fully-supervised performance on UCF101 and HMDB51, averaged over 3 splits. 'Ours': Video LA with IDT prior. DPC uses a non-standard version of 3D ResNet, and does not report fully-supervised performance for it. Our method shows top accuracy among the models using the same network architecture.

| Method | Network | Frame size | #Frames | UCF101 | | HMDB51 | |
|---|---|---|---|---|---|---|---|
| | | | | Acc. | % sup. | Acc. | % sup. |
| PMAS [43] | C3D | 112 × 112 | 16 | 61.2 | 74.3 | 33.4 | - |
| 3D-Puzzle [20] | 3D ResNet18 | 224 × 224 | 16 | 65.8 | 78.0 | 33.7 | 59.8 |
| DPC [13] | 3D ResNet18 | 112 × 112 | 40 | 68.2 | - | 34.5 | - |
| Ours | 3D ResNet18 | 112 × 112 | 16 | 73.0 | 86.5 | 41.6 | 73.8 |
| 3D-RotNet [19] | 3D ResNet18 | 112 × 112 | 64 | 66.0 | 72.1 | 37.1 | 55.5 |
| Ours | 3D ResNet18 | 112 × 112 | 64 | **83.0** | **90.7** | **50.4** | **75.6** |
| DPC [13] | 3D ResNet34 | 224 × 224 | 40 | 75.7 | - | 35.7 | - |
| CBT [35] | S3D | 112 × 112 | 16 | 79.5 | 82.1 | 44.6 | 58.8 |
| IDT [42] | - | Full | All | 85.9 | - | 57.2 | - |

IDT prior helping to better model the temporal information. Our approach also outperforms DPC [13], when the network depth is the same for both methods, even though DPC uses much longer sequences (40 frames with a stride 2, so the effective length is 120). Notably, on HMDB our approach even outperforms a variant of DPC with a deeper network, and bigger frame size by a large margin. When trained with longer temporal sequences, our method also outperforms the deeper variant of DPC on UCF by 7.3%. On HMDB we are 14.7% ahead.

The very recent approach of Sun et al. [35] ('CBT' in the table), reports high accuracy on both datasets. However, we show that this is due to the authors of [35] using a much deeper network than other methods in the literature. In terms of the fraction of fully-supervised performance, the 16-frame variant of our method outperforms CBT by 4.4% on UCF and by 15.0% on HMDB. Moreover, the 64-frame variant also outperforms CBT in raw accuracy on both datasets.

Finally, we report the performance of Fisher vector encoded IDT descriptors ('IDT' in the table, the numbers are taken from [32]). Please note that these descriptors are computed on the full length of the video, using the original resolution. Despite this, our 64-frame model comes close to the IDT performance.

## 5    Conclusions

This paper introduced a novel approach for unsupervised video representation learning. Our method transfers the heuristic-based IDT descriptors, that are effective at capturing motion information, to 3D ConvNets via non-parametric clustering. We quantitatively evaluated the learned representations on UCF101 and HMDB51, and demonstrated that they outperform prior work. We also qualitatively analyzed the discovered video clusters, showing that they successfully capture video dynamics, in addition to appearance. This analysis highlighted that the clusters do not perfectly match with the human-defined action classes, partially explaining the remaining gap to the fully-supervised performance.

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
