# OpenReview forum: "Unsupervised Learning of Video Representations via Dense Trajectory Clustering"
_thecvf.com/ECCV/2020/Workshop/VIPriors — VIPriors Oral_

### Official Review · AnonReviewer2 · 2020-07-27
**Unsupervised Learning of Video Representations via Dense Trajectory Clustering**

**Confidence:** 5
**Rating:** 9

**Review:**

#### 1. [Summary] In 2-3 sentences, describe the key ideas, experiments, and their significance.
The paper proposes a method which uses IDT descriptor and 3DConvNet to obtain action clusters and learns the unsupervised video representation.

#### 2. [Strengths] What are the strengths of the paper? Clearly explain why these aspects of the paper are valuable.
- Using IDT as prior knowledge
- Effective motion capturing
- Performance
- Extensive related works and ablation study

#### 3. [Weaknesses] What are the weaknesses of the paper? Clearly explain why these aspects of the paper are weak.
- It requires training with other big dataset.
- Do the authors have any evidence of the performance without the usage of any other dataset?

#### 4. [Overall rating] Paper rating
9

#### 5. [Justification of rating] Please explain how the strengths and weaknesses aforementioned were weighed in for the rating.
The paper has nice analyses and the proposed method outperforms other methods by using IDT descriptors and 3DConvNet.

#### 6. [Detailed comments] Additional comments regarding the paper (e.g. typos or other possible improvements you would like to see for the camera-ready version of the paper, if any.)
- L.318: Can you elaborate with the fine-tuning stage?
- Can you please clarify for the Table 3 if you train your network with Kinetics-400 or 600? Most of the methods (DPC, 3D-Puzzle, PMAS) in the table use Kinetics-400 for self supervised training.
- What are the limitations?
- Do you have any memory usage and time analyses?

Typos:
- L.406: first
- L.507: Kinetics

---

### Official Review · AnonReviewer1 · 2020-07-27
**Unsupervised Learning of Video Representations via Dense Trajectory Clustering**

**Confidence:** 3
**Rating:** 6

**Review:**

1. [Summary] In 2-3 sentences, describe the key ideas, experiments, and their significance.

This paper addresses the task of unsupervised learning of video representations for action recognition. Following current trend for image representation learning, authors propose first to adapt [46] and [49] for video instance recognition and local aggregation respectively. Since results prove that these methods do not capture motion, which is clearly important for action recognition, they proposed to force a 3D ConvNet to learn embeddings from IDTs. Experiment results justifies their framework.

2. [Strengths] What are the strengths of the paper? Clearly explain why these aspects of the paper are valuable.

The whole paper is well written and motivations are very well stated.
Although authors obtained promising results by correctly adapting [46] and [49], they went further and analyzed errors.

3. [Weaknesses] What are the weaknesses of the paper? Clearly explain why these aspects of the paper are weak.

A strong hypothesis of the paper is that motion is important to recognize actions in videos. And current 3D convnets models cannot learn it, while they tend to only learn appearance. For this reason, authors use IDTs.
This hypothesis seems unfair since 3D convs can be trained with flow, or even two-stream.
Have authors try their Video IR and Video LA using directly optical flow, or introducing a two-stream model? (C3D, I3D, R(2+1), TSN,…)

4. [Overall rating] Paper rating.

6

5. [Justification of rating] Please explain how the strengths and weaknesses aforementioned were weighed in for the rating.

Good research story, good experimental set-up but arguable assumption.

6. [Detailed comments] Additional comments regarding the paper (e.g. typos or other possible improvements you would like to see for the camera-ready version of the paper, if any.)

Do authors plan to release supplementary material they claim to have in the paper?

---

### Decision · Program_Chairs · 2020-07-29

**Decision:**

Accept (Oral)

**Comment:**

It is our pleasure to inform you that your paper has been accepted to the oral track of the 1st Visual Inductive Priors for Data-Efficient Deep Learning Workshop.

Please note the following deadlines:
* August 11, 2020 - workshop material, including:
 * paper in PDF format;
 * pre-recorded video presentation;
 * slides of the presentation in PDF.
* September 15, 2020 - camera-ready paper

The reviews can be found on OpenReview. Please take these comments and suggestions into account when preparing the camera-ready version of your paper, which is due September 15, 2020. The camera-ready paper should be uploaded to OpenReview.

As part of the workshop, each paper for oral presentation must submit a pre-recorded 5 minute talk before August 11, 2020. You will receive more information on how to upload the material shortly. The requirements for the video are:
* Duration: maximum 5 minutes
* MP4 format
* File size max. 100 MB
* Has an inset with a video of the speaker
* 16:9 aspect ratio (strongly preferred)
* 1920x1080 resolution (strongly preferred, at least 720 height)

Our suggested software for pre-recording your presentation is Zoom. For more information, please refer to the following guides:
How to record with Zoom Guide: http://homepages.inf.ed.ac.uk/rbf/ECCV2020HowtoRecordusingZoom.pdf
How to Record with Zoom tutorial: https://www.youtube.com/watch?v=CR199W7HdC0
Please ensure that at least one of the authors of the paper is available to attend the workshop during the allotted times. Note that the workshop will take place in two sessions spread across time zones (details are to follow). We will send instructions on how to connect to the workshop as soon as possible. The schedule for all talks and papers will be posted soon at the workshop website: https://vipriors.github.io.

We look forward to seeing you at the workshop!